# Error Correction Code Transformer

**Yoni Choukroun**
Tel Aviv University
choukroun.yoni@gmail.com

**Lior Wolf**
Tel Aviv University
liorwolf@gmail.com

## Abstract

Error correction code is a major part of the physical communication layer, ensuring the reliable transfer of data over noisy channels. Recently, neural decoders were shown to outperform classical decoding techniques. However, the existing neural approaches present strong overfitting, due to the exponential training complexity, or a restrictive inductive bias, due to reliance on Belief Propagation. Recently, Transformers have become methods of choice in many applications, thanks to their ability to represent complex interactions between elements. In this work, we propose to extend for the first time the Transformer architecture to the soft decoding of linear codes at arbitrary block lengths. We encode each channel's output dimension to a high dimension for better representation of the bits' information to be processed separately. The element-wise processing allows the analysis of channel output reliability, while the algebraic code and the interaction between the bits are inserted into the model via an adapted masked self-attention module. The proposed approach demonstrates the power and flexibility of Transformers and outperforms existing state-of-the-art neural decoders by large margins, at a fraction of their time complexity.

## 1   Introduction

Reliable digital communication is of major importance in the modern information age and involves the design of codes to be robustly decoded under noisy transmission channels. While optimal decoding is defined by the NP-hard maximum likelihood rule, the efficient decoding of algebraic block codes is an open problem. Recently, powerful learning-based techniques have been introduced to this field.

At present, deep learning models that implement parameterized versions of the legacy Belief Propagation (BP) decoders are dominant [19, 20, 18, 21, 4]. This approach is appealing, since the exponential size of the input space is reduced by ensuring that certain symmetry conditions are met. In this case, it is sufficient to train the network on a noisy version of the zero codeword. However, this approach remains extremely restrictive, due to the strong inductive bias induced by the Tanner graph.

Model-free decoders [23, 10, 13] employ variants of generic neural networks and may potentially benefit from the application of powerful architectures that have emerged in recent years in various fields. However, such decoders suffer from the curse of dimensionality, since the training grows exponentially with the number of information bits [31]. A major challenge of model-free decoders, therefore, is the difficulty of the network in learning the code and identifying its most reliable components, especially for the basic multi-layer perceptron architectures suggested in the literature.

In this work, we propose a *model-free* decoder built upon the Transformer architecture [29]. As far as we can ascertain, this is the first adaptation of Transformers to error correction codes. Our Transformer employs a high-dimensional *scaled* element-wise embedding of the input which can be seen as positional and reliability encoding. Our decoder employs an *adapted-mask* self-attention mechanism, in which the interaction between bits follows the code's parity check matrix, enabling the incorporation of domain knowledge into the model.

36th Conference on Neural Information Processing Systems (NeurIPS 2022).

Applied to a wide variety of codes, our method outperforms the state-of-the-art learning-based solutions by very large margins, even with very shallow architectures, enabling potential deployment. This is the first time a decoder designed de-novo outperforms neural BP-based decoders.

## 2  Related Works

Over the past few years, the emergence of deep learning has demonstrated the advantages of Neural Networks in many communication applications, such as channel equalization, modulation, detection, quantization, compression, and *decoding* [12]. In [23] the entire coding-decoding channel transmission pipeline was abstracted as a fully parameterized autoencoder. In many scenarios, neural network-based methods outperform existing decoders, as well as popular codes [20, 10].

Two main classes of Neural decoders can be established in the current literature [25]. Model-free decoders employ general types of neural network architectures, as in [5, 10, 13, 2]. The results obtained in [10] for polar codes ($n = 16$) are similar to the maximum a posteriori (MAP) decoding results. However, the exponential number of possible codewords make the decoding of larger codes unfeasible. In [2], a preprocessing of the channel output allows the decoder to remain provably invariant to the codeword and enables the deployment of any kind of model without any overfitting cost. The model-free approaches generally make use of stacked fully connected networks or recurrent neural networks in order to simulate the iterative process existing in many legacy decoders. However, these architectures have difficulties in learning the code and analyzing the reliability of the output, and generally require prohibitive parameterization or expensive graph permutation preprocessing [2].

The second class is that of model-based decoders [19, 20] implementing augmented parameterized versions of classical BP decoders, where the Tanner graph is unfolded into a neural network in which weights are assigned to each variable edge, resulting in improvement in comparison to the baseline BP method. This approach remains limited since it is built on top of a restrictive model. Recently, [25] presented a data-driven framework for codeword permutation selection as a pre-decoding stage. A self-attention mechanism is applied over several permutation-specific embeddings to estimate the best permutation (among a given set) to be decoded. A combination of the BP model with a hyper-graph network was shown to enhance performance [21]. Further improvements over the latter were obtained by introducing an autoregressive signal based on previous state conditioning and SNR estimation [22].

Transformer neural networks were originally introduced for machine translation [29] and now dominate the performance tables for most applications in the field of Natural Language Processing (NLP), e.g., [29, 7]. Transformer encoders primarily rely on the self-attention operation in conjunction with feed-forward layers, allowing manipulation of variable-size sequences and learning of long-range dependencies. Many works extended the Transformer architecture from NLP to various applications, such as computer vision [8], speech analysis [17] and molecular properties prediction [6].

## 3  Background

We provide the necessary background on error correction coding and the Transformer architecture.

### 3.1  Coding

We assume a standard transmission that uses a linear code $C$. The code is defined by the binary generator matrix $G$ of size $k \times n$ and the binary parity check matrix $H$ of size $(n - k) \times n$ defined such that $GH^T = 0$ over the two elements' Galois field.

The input message $m \in \{0, 1\}^k$ is encoded by $G$ to a codeword $x \in C \subset \{0, 1\}^n$ satisfying $Hx = 0$ and transmitted via a Binary-Input Symmetric-Output channel (e.g. an AWGN channel). Let $y$ denote the channel output represented as $y = x_s + z$, where $x_s$ denotes the Binary Phase Shift Keying (BPSK) modulation of $x$ (i.e., over $\{\pm 1\}$), and $z$ is random noise independent of the transmitted $x$.

The main goal of the decoder $f : \mathbb{R}^n \to \mathbb{R}^n$ is to provide a soft approximation $\hat{x} = f(y)$ of the codeword. An illustration of the coding framework is presented in Figure 1.

We wish to employ the Transformer architecture as part of a model-free approach. We, therefore, follow the preprocessing and post-processing of [2] to remain provably invariant to the transmitted

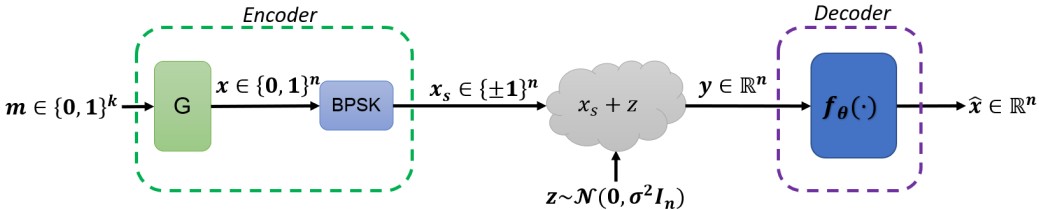

Figure 1: Illustration of the communication system. Our work focuses on the design and training of the parameterized decoder $f_\theta$.

codeword and to avoid overfitting. This step involves a transformation of the channel output $y$, which entails a loss of information without implying any intrinsic performance penalty in decoding, i.e., the preprocessing coupled with an appropriately designed decoder can achieve MMSE decoding.

The preprocessing replaces $y$ with a vector of dimensionality $2n - k$ defined as

$$\tilde{y} = h(y) = [|y|, s(y)],  \tag{1}$$

where, $[\cdot, \cdot]$ denotes vector concatenation, $|y|$ denotes the absolute value (magnitude) of $y$ and $s(y) = Hy_b \in \{0, 1\}^{n-k}$ denotes the binary code *syndrome* obtained via the multiplication of the parity check matrix by the binary mapping $y_b$ of $y$ such that $y_b = 0.5(1 - \mathrm{sign}(y))$.

The post-processing predicts the transmitted codeword by multiplying the elements of $y$ with the predicted multiplicative noise $\tilde{z}$ defined such that $y = x_s\tilde{z}$. Namely, the prediction takes the form

$$\hat{x} = y \cdot f(h(y)).  \tag{2}$$

In our setting, the decoder is parameterized by $\theta$ such that $\hat{x} = y \cdot f_\theta(|y|, Hy_b)$.

## 3.2 Transformers

The Transformer was introduced as a novel, attention-based building block for machine translation [29]. The input sequence is first embedded into a high-dimensional space, coupled with positional embedding for each element. The embeddings are then propagated through multiple normalized self-attention and feed-forward blocks.

The self-attention mechanism introduced by Transformers is based on a trainable associative memory with (key, value) vector pairs, where a query vector $q \in \mathbb{R}^d$ is matched against a set of $k$ key vectors using scaled inner products, as follows

$$A(Q, K, V) = \mathrm{Softmax}\left(\frac{QK^T}{\sqrt{d}}\right)V,  \tag{3}$$

where $Q \in \mathbb{R}^{N \times d}$, $K \in \mathbb{R}^{k \times d}$ and $V \in \mathbb{R}^{k \times d}$ represent the packed $N$ queries, $k$ keys and values tensors respectively. Keys, queries and values are obtained using linear transformations of the sequence's elements. A multi-head self-attention layer is defined by extending the self-attention mechanism using $h$ attention *heads*, i.e. $h$ self-attention functions applied to the input, reprojected to values via a $dh \times D$ linear layer.

# 4  Error Correction Code Transformer

We present the elements of the proposed Transformer decoder, the complete architecture, and the training procedure.

## 4.1  Positional Reliability Encoding

We first consider each dimension of $\{\tilde{y}_i\}_{i=1}^{2n-k}$ separately and project each one to a high $d$ dimensional embedding $\{\phi_i\}_{i=1}^{2n-k}$ such that

$$\phi_i = \begin{cases} |y_i|W_i, & \text{if } i \leq n \\ \left(1 - 2(s(y))_{i-n+1}\right)W_i, & \text{otherwise} \end{cases}  \tag{4}$$

where $\{W_j \in \mathbb{R}^d\}_{j=1}^{2n-k}$ denotes the one-hot encoding defined according to the bit position. The embedding is modulated by the magnitude and syndrome values, such that less reliable elements (i.e. low magnitude) collapse to the origin. This property becomes especially appealing when applied to the standard dot product of the self-attention module: for the first layer and two distinct information bits embedding $\phi_i, \phi_j$ we have

$$\langle \phi_i, \phi_j \rangle = \begin{cases} |y_i||y_j|\langle W_i, W_j \rangle & i,j \leq n \\ |y_i|(1 - 2(s(y))_{j-n+1})\langle W_i, W_j \rangle, & i \leq n < j, \end{cases} \tag{5}$$

where an unreliable information bit vanishes and a non-zero syndrome entails a negative scaling, potentially reducing the impact on the softmax aggregation, as illustrated in Sec. 6.

In contrast to [2], which requires permutation of the code in order to artificially provide to the fully connected network indications about the most reliable channel outputs, in our construction, the channel output reliability is directly obtained and maintained in the network via the scaled bit-wise embedding. The proposed scaled encoding can be thought of as a *positional* encoding according to the input *reliability* since the bit positions are fixed. We note that representing the codeword by a set of one-hot vectors is very different from any of the existing methods.

### 4.2 Code-Aware Self-Attention

In order to detect and correct errors, a decoder must *analyze and compare* the received word bits via the parity check equations, such that a non-zero syndrome indicates that channel errors have occurred. In our Transformer-based architecture, bit comparisons become natural via the self-attention *interaction* mechanism.

However, comparing every pair of elements, as is generally performed in Transformer architectures, is sub-optimal, since not every bit is necessarily related to all the others. We propose to make use of the self-attention mechanism in order to incorporate fundamental domain knowledge about the relevant code. Specifically, we use it to indicate that the syndrome values should only be dependent on the corresponding parity check bits.

Thus, for a given algebraic code defined by the matrix $H$ we propose to define a function $g(H) : \{0,1\}^{(n-k) \times k} \to \{-\infty, 0\}^{2n-k \times 2n-k}$ defining the mask to be applied to the self-attention mechanism, such that

$$A_H(Q, K, V) = \text{Softmax}\left(\frac{QK^T + g(H)}{\sqrt{d}}\right)V. \tag{6}$$

We propose to build the *symmetric* mask such that it contains information about *every* pairwise bit relation, as follows. The mask is first initialized as the identity matrix. For each row $i$ of the binary parity check matrix $H$, we unmask at the locations of *every pair* of ones in the row, since those bits are connected and may impact each other as well as the syndrome in the decoding procedure. We also unmask the location of pairs of ones with the corresponding syndrome bit at $n + i$, since they define the parity check equations. Illustrations of typical self-attention maps are given in Appendix A.

---

**Algorithm 1:** Mask construction Pseudo Code

---

```
1  function g(H)
2      l,n = H.shape
3      k = n-l
4      mask = eye(2n-k)
5      for i in range(0,n-k) do
6          idx = where(H[i]==1)
7          for j in idx do
8              mask[n+i,j] = mask[j,n+i] = 1
9              for k in idx do
10                 mask[j,k] = mask[k,j] = 1

11     return -∞(¬ mask)
```

---

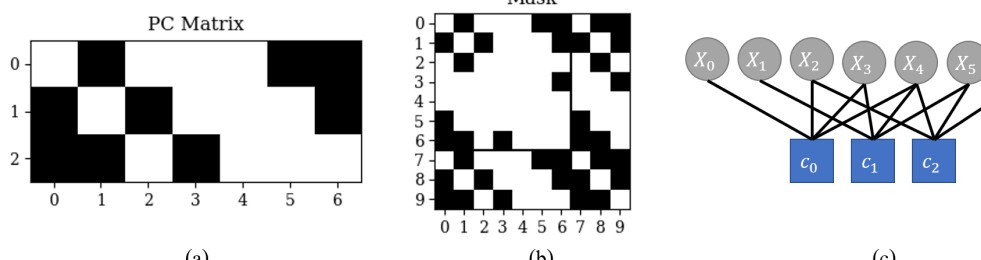

(a)                     (b)                     (c)

Figure 2: We present the corresponding parity-check matrix (a), the induced Tanner graph (c) and the proposed masking (b) on the Hamming(7,4) code.

We summarize the construction of the mask in Algorithm 1. This construction enables greater freedom in decoding than the relations enabled by the Tanner graph, since related bits may impact each other *beyond* the parity check equations as depicted in Figure 2. While regular Transformers can be assimilated to a neural network applied on a complete graph, the proposed mask can be seen as the adjacency matrix of the Tanner graph extended to a two rings connectivity. In contrast to BP, which collapses the information via interleaved variable and check layers, the masked self-attention allows the simultaneous cross-analysis of related elements.

Most importantly, since the mask is fixed and computed once, the self-attention quadratic complexity bottleneck $\mathcal{O}(n^2 d)$ is now reduced to the density of the code $\mathcal{O}(\sum_{ij}(H)_{ij}d)$. This property is especially appealing for low-density codes (e.g. [9]), while in all our experiments the original complexity of the self-attention is reduced by $84\%$ on average, as presented in Sec. 6.

### 4.3 Architecture and Training

The initial encoding is defined as a $d$ dimensional one hot encoding of the $2n - k$ input elements. The decoder is defined as a concatenation of $N$ decoding layers composed of self-attention and feed-forward layers interleaved by normalization layers. The output module is defined by two fully connected layers. The first layer reduces the element-wise embedding to a one-dimensional $2n - k$ vector and the second to a $n$ dimensional vector representing the soft decoded noise. An illustration of the model is given in Figure 3.

The dimension of the feed-forward network is four times that of the embedding [29] and is composed of GEGLU layers [27], with layer normalization set to the pre-layer norm setting, as in [15, 32]. We use an eight-head self-attention module in all experiments. colorblue The impact of the number of heads on the accuracy is further explored in Appendix C.

The training objective is the cross-entropy function where the goal is to learn to predict the *multiplicative* noise $\tilde{z}$ [2]. Denoting the *soft* multiplicative noise $\tilde{z}_s$ such that $y = x_s \tilde{z}_s$, we obtain $\tilde{z}_s = \tilde{z}_s x_s^2 = y x_s$. Thus, the binary multiplicative noise to be predicted is defined by $\tilde{z} = \text{bin}(y \cdot x_s)$, such that the loss computed for a single received word $y$ is

$$\mathcal{L} = -\sum_{i=1}^{n} \tilde{z}_i \log(f_\theta(y)) + (1 - \tilde{z}_i) \log(1 - f_\theta(y)). \tag{7}$$

The estimated hard-decoded codeword is straightforwardly obtained as $\hat{x}_b = \text{bin}(\text{sign}(f_\theta(y) \cdot y))$.

The Adam optimizer [14] is used with 128 samples per minibatch, for 1000 epochs, with 1000 minibatches per epoch. For $N = 10$ architectures we trained the models for 1500 epochs. We note that using more epochs can improve performance. However, the current setting is already more than enough to reach new SOTA performance. Due to the construction of the model and its input preprocessing, the zero codeword is enough for training. The additive Gaussian noise is sampled randomly per batch in the $\{3, \ldots, 7\}$ normalized SNR (i.e. $E_b/N_0$) range. Additional experiments with the non-Gaussian Rayleigh fading channel are provided in Appendix E. We initialized the learning rate to $10^{-4}$ coupled with a cosine decay scheduler down to $5 \cdot 10^{-7}$ at the end of the training. No warmup was employed [32]. Training time range from 12 to 24 hours depending on the code length, and no optimization of the self-attention mechanism have been employed. Training and

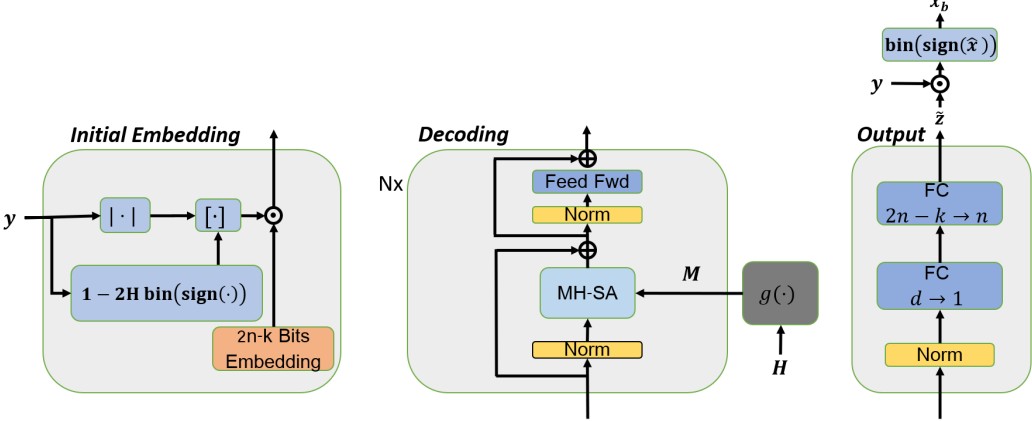

Figure 3: Illustration of the proposed Transformer architecture. The main differences from other Transformers are the initial scaled bit-embedding, the construction of the code-aware masked self-attention, and the output module.

experiments have been performed on a 12GB Titan V GPU. The training time was in the range of 19-40, 40-102, and 56-171 seconds per epoch for the $N = 2, 6, 10$ architectures, respectively.

# 5   Experiments[1]

To evaluate our method, we train the proposed architecture with three classes of linear block codes: Low-Density Parity Check (LDPC) codes [9], Polar codes [1] and Bose–Chaudhuri–Hocquenghem (BCH) codes [3]. All parity check matrices are taken from [11]. The proposed architecture is defined solely by the number of encoder layers $N$ and the dimension of the embedding $d$. We compare our method to the BP algorithm, to the augmented hypernetwork BP algorithm of [21] (hyp BP), to the RNN architecture of [2] and to the very recent SOTA performance of [22] (AR BP). All the results were obtained from the corresponding papers. A comparison to the SCL Polar decoder [28] is given in Appendix D. The results are reported as bit error rates (BER) for different normalized SNR values (dB). We follow the testing benchmark of [21, 22]. During testing, our decoder decodes at least $10^5$ random codewords, to obtain at least $500$ frames with errors at each SNR value.

The results are reported in Tab. 1, where we present the negative natural logarithm of the BER. For each code we present the results of the BP-based competing methods for 5 and 50 iterations (first and second rows), corresponding to a neural network with 10 and 100 layers, respectively. We present our framework performance for six different architectures with $N = \{2, 6\}$ $d = \{32, 64, 128\}$ respectively (first to third rows). Since LDPC codes are optimized for BP decoding [26], we also present for these codes the performance of a $N = 10, d = 128$ architecture. As can be seen, our approach outperforms current SOTA results by very large margins on several codes, at a fraction of the number of iterations, and is able to outperform legacy methods (e.g. BP) *at convergence*, even with extremely shallow architectures (e.g. $N = 2, d = 32$). We remark that the proposed method seems to perform better for high rate (i.e. $\frac{k}{n}$) codes.

We provide plots of more BER values for some of the codes in Figure 4(a,b) for Polar(64,32) and BCH(63,51) codes, respectively. Our method is able to outperform the converged AR-BP by up to two orders of magnitude for high SNR values. Performance on larger codes is studied in Appendix H.

In Fig. 4(c), We compare our method with the best model of [2] on the only code provided BCH(127,64). Their network is built as a five-layer stacked GRU model, with five iterations with an embedding of $5(2n - k) = 950$ trained for 5000 epochs, with a batch size of 1000. As can be seen, our method outperforms this model by up to $211\%$, with only half the number of layers, one-tenth as many parameters, and a fraction of the training requirements.

---

[1]Code available at `https://github.com/yoniLc/ECCT`

Table 1: A comparison of the negative natural logarithm of Bit Error Rate (BER) for three normalized SNR values of our method with literature baselines. Higher is better.

Concurrent results are obtained after $L = 5$ BP iterations in first row (i.e. 10 layers neural network) and *at convergence* results in second row obtained after $L = 50$ BP iterations (i.e. 100 layers neural network). Best results in **bold**, second best is underlined, and the minimal Transformer architecture to outperform every other competing method is in *italic*.

Our performance is presented for seven different architectures: for $N = \{2, 6\}$, we present results for $d = \{32, 64, 128\}$ (first to third rows), and for $N = 10$ we run only the $d = 128$ configuration.

| Method | BP[24] | | | Hyp BP[21] | | | AR BP[22] | | | Ours N=2 | | | Ours N=6 | | | Ours N=10 | | |
|---|---|---|---|---|---|---|---|---|---|---|---|---|---|---|---|---|---|---|
| | 4 | 5 | 6 | 4 | 5 | 6 | 4 | 5 | 6 | 4 | 5 | 6 | 4 | 5 | 6 | 4 | 5 | 6 |
| Polar(64,32) | 3.52 | 4.04 | 4.48 | 4.25 | 5.49 | 7.02 | 4.77 | 6.30 | 8.19 | 4.27 | 5.44 | 6.95 | *5.71* | *7.63* | *9.94* | | | |
| | 4.26 | 5.38 | 6.50 | 4.59 | 6.10 | 7.69 | 5.57 | 7.43 | 9.82 | 4.57 | 5.86 | 7.50 | 6.48 | 8.60 | 11.43 | | | |
| | | | | | | | | | | 4.87 | 6.2 | 7.93 | **6.99** | **9.44** | **12.32** | | | |
| Polar(64,48) | 4.15 | 4.68 | 5.31 | 4.91 | 6.48 | 8.41 | 5.25 | 6.96 | 9.00 | 4.92 | 6.46 | 8.41 | *5.82* | *7.81* | 10.24 | | | |
| | 4.74 | 5.94 | 7.42 | 4.92 | 6.44 | 8.39 | 5.41 | 7.19 | 9.30 | 5.14 | 6.78 | 8.9 | 6.15 | 8.20 | 10.86 | | | |
| | | | | | | | | | | 5.36 | 7.12 | *9.39* | **6.36** | **8.46** | **11.09** | | | |
| Polar(128,64) | 3.38 | 3.80 | 4.15 | 3.89 | 5.18 | 6.94 | 4.02 | 5.48 | 7.55 | 3.51 | 4.52 | 5.93 | 4.47 | 6.34 | 8.89 | | | |
| | 4.10 | 5.11 | 6.15 | 4.52 | 6.12 | 8.25 | 4.84 | 6.78 | 9.30 | 3.83 | 5.16 | 7.04 | *5.12* | *7.36* | *10.48* | | | |
| | | | | | | | | | | 4.04 | 5.52 | 7.62 | **5.92** | **8.64** | **12.18** | | | |
| Polar(128,86) | 3.80 | 4.19 | 4.62 | 4.57 | 6.18 | 8.27 | 4.81 | 6.57 | 9.04 | 4.30 | 5.58 | 7.34 | 5.36 | *7.45* | *10.22* | | | |
| | 4.49 | 5.65 | 6.97 | 4.95 | 6.84 | 9.28 | 5.39 | 7.37 | 10.13 | 4.49 | 5.90 | 7.75 | 5.75 | 8.16 | 11.29 | | | |
| | | | | | | | | | | 4.75 | 6.25 | 8.29 | **6.31** | **9.01** | **12.45** | | | |
| Polar(128,96) | 3.99 | 4.41 | 4.78 | 4.73 | 6.39 | 8.57 | 4.92 | 6.73 | 9.30 | 4.56 | 5.98 | 7.93 | *5.39* | *7.62* | 10.45 | | | |
| | 4.61 | 5.79 | 7.08 | 4.94 | 6.76 | 9.09 | 5.27 | 7.44 | 10.2 | 4.69 | 6.20 | 8.30 | 5.88 | 8.33 | 11.49 | | | |
| | | | | | | | | | | 4.88 | 6.58 | 8.93 | **6.31** | **9.12** | **12.47** | | | |
| LDPC(49,24) | 5.30 | 7.28 | 9.88 | 5.76 | 7.90 | 11.17 | 6.05 | 8.13 | 11.68 | 4.51 | 6.07 | 8.11 | 5.74 | 8.13 | 11.30 | ***6.68*** | ***9.53*** | ***13.30*** |
| | 6.23 | 8.19 | 11.72 | 6.23 | 8.54 | 11.95 | 6.58 | 9.39 | 12.39 | 4.58 | 6.18 | 8.46 | 5.91 | 8.42 | 11.90 | | | |
| | | | | | | | | | | 4.71 | 6.38 | 8.73 | 6.13 | 8.71 | 12.10 | | | |
| LDPC(121,60) | 4.82 | 7.21 | 10.87 | 5.22 | 8.29 | 13.00 | 5.22 | 8.31 | 13.07 | 3.88 | 5.51 | 8.06 | 4.98 | 7.91 | 12.70 | ***5.73*** | ***9.35*** | ***15.01*** |
| | - | - | - | - | - | - | - | - | - | 3.89 | 5.55 | 8.16 | 5.02 | 7.94 | 12.72 | | | |
| | | | | | | | | | | 3.93 | 5.66 | 8.51 | 5.17 | 8.31 | 13.30 | | | |
| LDPC(121,70) | 5.88 | 8.76 | 13.04 | 6.39 | 9.81 | 14.04 | 6.45 | 10.01 | 14.77 | 4.63 | 6.68 | 9.73 | 6.11 | 9.62 | 15.10 | *7.13* | **11.50** | **17.92** |
| | - | - | - | - | - | - | - | - | - | 4.64 | 6.71 | 9.77 | 6.28 | 10.12 | 15.57 | | | |
| | | | | | | | | | | 4.67 | 6.79 | 9.98 | 6.40 | *10.21* | *16.11* | | | |
| LDPC(121,80) | 6.66 | 9.82 | 13.98 | 6.95 | 10.68 | 15.80 | 7.22 | 11.03 | 15.90 | 5.27 | 7.59 | 10.08 | 6.92 | 10.74 | 15.10 | **8.12** | **12.90** | 17.82 |
| | - | - | - | - | - | - | - | - | - | 5.29 | 7.63 | 10.90 | 7.17 | *11.21* | *16.31* | | | |
| | | | | | | | | | | 5.30 | 7.65 | 11.03 | *7.41* | 11.51 | 16.44 | | | |
| MacKay(96,48) | 6.84 | 9.40 | 12.57 | 7.19 | 10.02 | 13.16 | 7.43 | 10.65 | 14.65 | 4.95 | 6.67 | 8.94 | 6.88 | 9.86 | 13.40 | *8.39* | **12.24** | **16.41** |
| | - | - | - | - | - | - | - | - | - | 5.04 | 6.80 | 9.23 | 7.10 | 10.12 | 14.21 | | | |
| | | | | | | | | | | 5.17 | 7.07 | 9.64 | 7.38 | *10.72* | *14.83* | | | |
| CCSDS(128,64) | 6.55 | 9.65 | 13.78 | 6.99 | 10.57 | 15.27 | 7.25 | 10.99 | 16.36 | 4.35 | 6.01 | 8.30 | 6.34 | 9.80 | 14.40 | *8.02* | **12.60** | **17.75** |
| | - | - | - | - | - | - | - | - | - | 4.41 | 6.09 | 8.49 | 6.65 | 10.40 | 15.46 | | | |
| | | | | | | | | | | 4.59 | 6.42 | 9.02 | 6.88 | 10.90 | 15.90 | | | |
| BCH(31,16) | 4.63 | 5.88 | 7.60 | 5.05 | 6.64 | 8.80 | 5.48 | 7.37 | 9.61 | 4.51 | 5.74 | 7.35 | *5.74* | *7.42* | *9.59* | | | |
| | - | - | - | - | - | - | - | - | - | 4.78 | 6.15 | 7.98 | 5.85 | 7.52 | 10.08 | | | |
| | | | | | | | | | | 5.18 | 6.82 | 8.91 | **6.39** | **8.29** | **10.66** | | | |
| BCH(63,36) | 3.72 | 4.65 | 5.66 | 3.96 | 5.35 | 7.20 | 4.33 | 5.94 | 8.21 | 3.79 | 4.87 | 6.35 | 4.42 | 5.91 | 8.01 | | | |
| | 4.03 | 5.42 | 7.26 | 4.29 | 5.91 | 8.01 | 4.57 | 6.39 | 8.92 | 4.05 | 5.28 | 7.01 | *4.62* | 6.24 | 8.44 | | | |
| | | | | | | | | | | 4.21 | 5.50 | 7.25 | **4.86** | ***6.65*** | ***9.10*** | | | |
| BCH(63,45) | 4.08 | 4.96 | 6.07 | 4.48 | 6.07 | 8.45 | 4.80 | 6.43 | 8.69 | 4.47 | 5.88 | 7.81 | *5.16* | *7.02* | *9.75* | | | |
| | 4.36 | 5.55 | 7.26 | 4.64 | 6.27 | 8.51 | 4.97 | 6.90 | 9.41 | 4.66 | 6.16 | 8.17 | 5.41 | 7.49 | 10.25 | | | |
| | | | | | | | | | | 4.79 | 6.39 | 8.49 | **5.60** | **7.79** | **10.93** | | | |
| BCH(63,51) | 4.34 | 5.29 | 6.35 | 4.64 | 6.08 | 8.16 | 4.95 | 6.69 | 9.18 | 4.60 | 6.05 | 8.05 | *5.20* | 7.08 | 9.65 | | | |
| | 4.5 | 5.82 | 7.42 | 4.80 | 6.44 | 8.58 | 5.17 | 7.16 | 9.53 | 4.78 | 6.34 | 8.49 | 5.46 | 7.57 | 10.51 | | | |
| | | | | | | | | | | 5.01 | 6.72 | 9.03 | **5.66** | **7.89** | **11.01** | | | |

# 6 Analysis

We study the impact of the proposed embedding and the masking procedure; furthermore, we analyze and compare the complexity of our method with other methods.

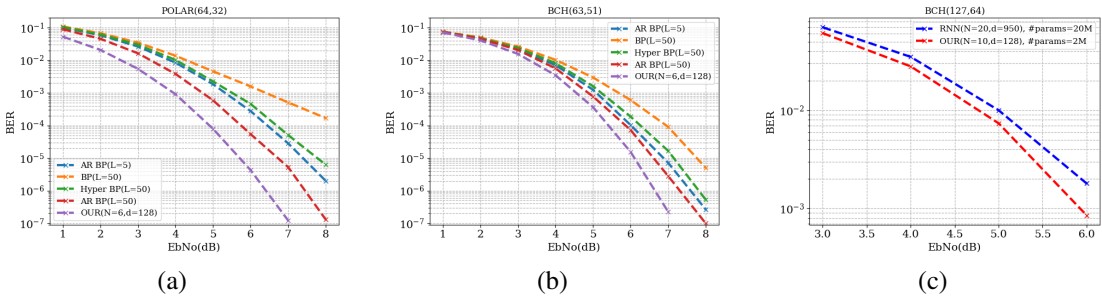

Figure 4: BER for various values of SNR for (a) Polar(64,32) and (b) BCH(63,51) codes. Comparison with the best model of Bennatan et al. [2] on BCH(127,64) code (c).

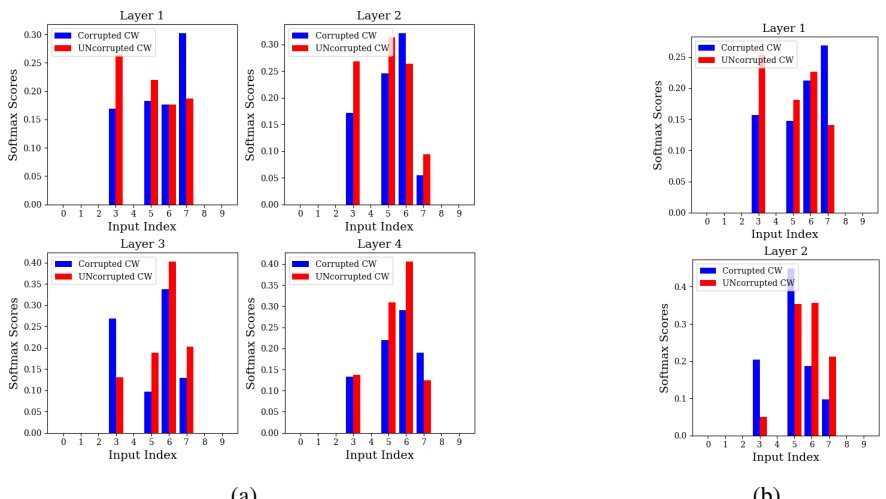

Figure 5: Analysis of the scaled embedding values of the first column of the masked self-attention map with corrupted and uncorrupted codeword (CW) for the standardized Hamming code, using the (a) $(N, d) = (4, 32)$ and (b) $(N, d) = (2, 32)$ architecture. The impact of the noisy embedding on the self-attention's first parity-check element (index 7) is visible.

## 6.1 Impact of the Reliability Embedding

We depict in Figure 5 the impact of the *scaled* embedding on the self-attention mechanism. We chose the popular Hamming$(7, 4)$ code, corrupting the zero codeword with additive noise at the *first* bit (zero bit index), involving a non-zero syndrome at the first parity check equation. We present the *masked* softmax values of the self-attention map at index zero, i.e. we visualize the influence of the first (unreliable) bit on the self-attention aggregation (first column of the map). As can be seen, the first bit embedding is impactless when it is corrupted and then detectable, while its impact on the syndrome embedding is considerably increased (index 7). Once the NN corrects the bit (last layer(s)), the values return to normal. A similar analysis for larger codes can be found in in Appendix B.

## 6.2 Ablation Study: Masking

We present in Figure 6 the impact of the proposed mask on the convergence. We show an *unmasked* architecture, where we leave the network to learn the code by itself (i.e. $g(H) = 0$), and compare it with our proposed masking approach. The benefit obtained with the mask is clear and understandable in light of the difficulties the Transformer may have in learning the code by itself. Therefore, the connectivity provided by the proposed masking framework reduces the loss by 76%, 69% and 66% for the BCH$(63, 36)$, POLAR$(64, 32)$ and LDPC$(49, 24)$ codes, respectively.

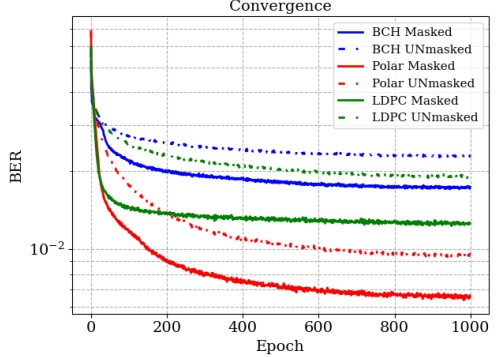

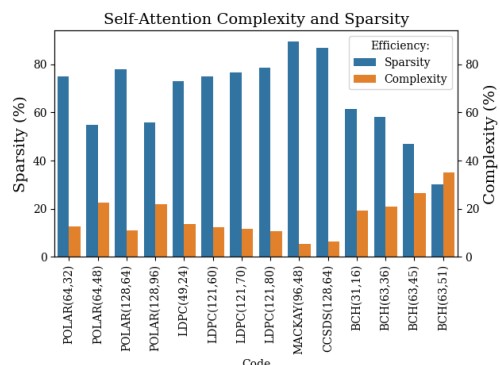

Figure 6: The effect of the proposed masking procedure on the convergence for $BCH(63, 36)$, $POLAR(64, 32)$ and $LDPC(49, 24)$ using $(N, d) = (6, 64)$.

Figure 7: Sparsity and complexity ratio of the proposed self-attention map for several codes.

## 6.3 Complexity Analysis

We first present, in Figure 7, the sparsity ratio of the *masked* self-attention map, as well as the induced complexity ratio of the *symmetric* map for several codes presented in the Results section. The ratio is computed with respect to the $\mathcal{O}(n^2)$ legacy self-attention map size. We can see that sparsity can reach up to more than $80\%$ while the computation ratios range from 5 to $35\%$ only. This improvement can lead to a sizable reduction in power and computation time on a *dedicated* hardware.

The complexity of the network is defined by $\mathcal{O}(N(d^2(2n - k) + hd)$, where $h \ll n^2$ denotes the *fixed* number of computations of the self-attention module. The real-world complexity for different codes is explored in Appendix G. As with most neural decoding approaches, while being extremely effective, the proposed method may remain inefficient compared to non-learning solutions, such as [24], in terms of memory requirement, power consumption, and computational resources, limiting their deployment potential. Acceleration of the proposed Transformer via architectural modification or NN acceleration (e.g. pruning, quantization) [30, 16] is left for future work.

In comparison, the existing AR-BP SOTA method of [22] requires $\mathcal{O}(2N(nd_v d_f d_g + n^2 d_v + (n - k)^2 + d_f))$ more operations than Hyper BP[21], where $d_v, d_f, d_g$ are the number of variable nodes, the capacity of the hypernetwork and primary network, respectively. Typical networks $f$ and $g$ are a 128 dimensional FC network with 4 layers and a 16 dimensional FC network with 2 layers respectively. With these hyperparameters, the $nd_v d_f d_g$ part of the complexity is approximately $nd_v 128^2$. Moreover, the method of [22] cannot be parallelized, due to the hypernetwork structure, i.e. each sample defines a different set of parameters. Thus, our network far surpasses the existing state-of-the-art error rates, with much less time complexity. Hyper-BP[21] is slightly less computationally intensive, since it adds a complexity term of $\mathcal{O}\left(2Nnd_v d_f d_g\right)$ on top of BP, but this approach as well as the original model-based network of [19] are far less competitive in terms of accuracy.

## 7 Conclusions

We present a novel Transformer architecture for decoding algebraic block codes. The proposed model allows effective representation of interactions, based on the high-dimensional embedding of the channel output and a code-dependent masking scheme of the self-attention module. The proposed framework is more efficient than standard Transformers, due to the sparse design of existing block codes, making its training and deployment extremely affordable. Even with a limited number of computational blocks, it outperforms popular (neural) decoders on a broad range of code families. We believe this new architecture will allow the development and adoption of new families of performing Transformer-based codes capable of setting new standards in this field.

## Acknowledgments

This project has received funding from the European Research Council (ERC) under the European Union's Horizon 2020 research and innovation programme (grant ERC CoG 725974). The contribution of the first author is part of a PhD thesis research conducted at Tel Aviv University.

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
