# Error Correction Code Transformer

**Yoni Choukroun**
Tel Aviv University
choukroun.yoni@gmail.com

**Lior Wolf**
Tel Aviv University
liorwolf@gmail.com

## A    Self-Attention Maps Visualization

In this section, we present the self-attention maps for various codes obtained at the $E_b/N_0 = 5$ noise level. Figure 1, presents the self-attention maps and the corresponding input for the Hamming BCH(7,4) code with $N = 6, d = 32$, similarly to the results presented in Section 6.1.

For better visual analysis we present in Figure 2 the self-attention maps for two longer codes. Interestingly, in the early stage of the decoding, ECCT seems to focus its processing of the syndrome. At deeper layers towards the final prediction, the focus shifts to the information bits.

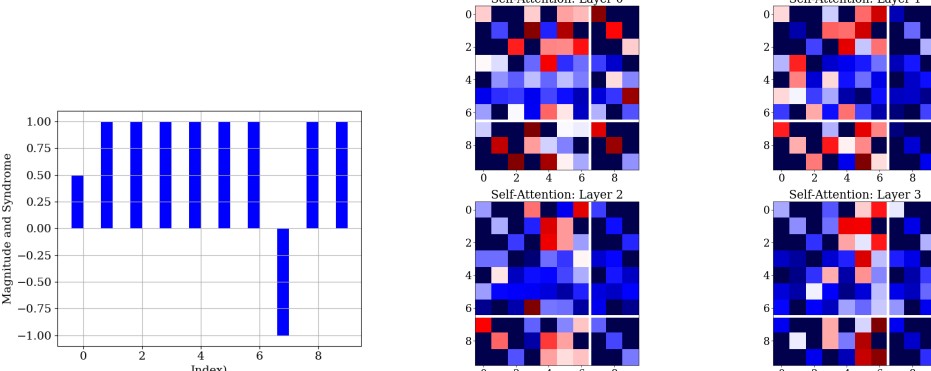

Figure 1: Illustration of the input $h(y)$ (magnitude and syndrome concatenation) and of the self-attention maps at different layers of the network, for the Hamming BCH(7,4) code with $N = 6, d = 32$. The self-attention heads are averaged and the white lines denote the separation between the magnitude and syndrome elements.

## B    Impact of the Reliability Embedding

Similarly to Section 6.1, we present in Figure 3 the impact of the reliability embedding on the self-attention map on a larger code. We chose the BCH$(31, 16)$ code which is still short enough for providing a clear visualization, and similarly to Section 6.1, the zero codeword is corrupted with additive noise at the first bit (zero bit index), involving a non-zero syndrome at the first parity check equation (i.e. the 32-th element of the embedding). As can be seen, the first bit embedding is impact-less when it is corrupted and then detectable, while its impact on the syndrome embedding is considerably increased. Once the network corrects the bit (last layers(s)), the values return to normal.

36th Conference on Neural Information Processing Systems (NeurIPS 2022).

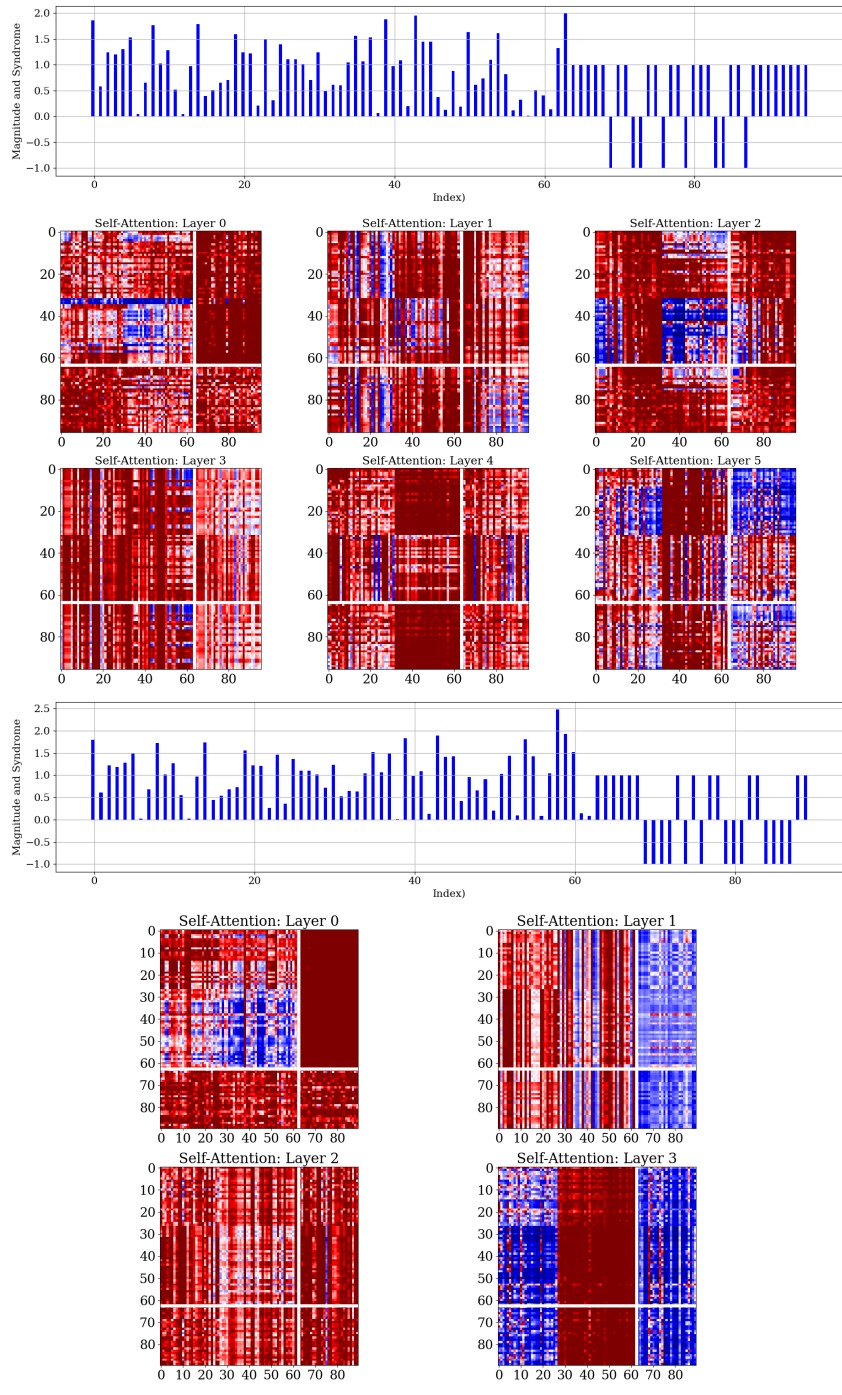

Figure 2: Illustration of the input $h(y)$ (magnitude and syndrome concatenation) and of the self-attention maps at different layers of the network, for POLAR(62,32) code with $N = 6, d = 32$ (top) and BCH(63,36) code with $N = 4, d = 32$ (bottom). The self-attention heads are averaged and the white lines denote the separation between the magnitude and syndrome elements. One can observe in the first layers that high attention values are assigned to the syndrome part.

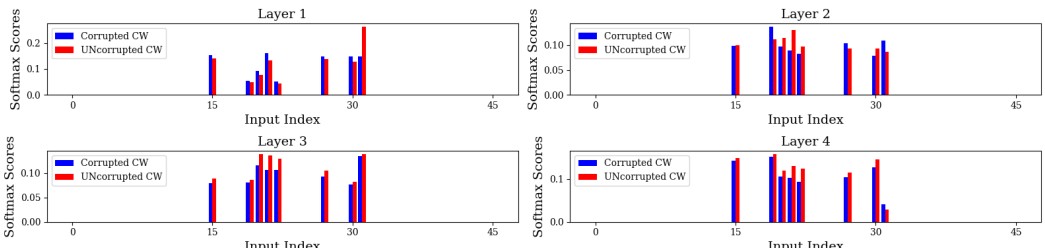

Figure 3: Analysis of the scaled embedding values of the first column of the masked self-attention map with corrupted and uncorrupted codeword (CW) for the standardized BCH$(31, 16)$ code, using the $(N, d) = (4, 32)$ architecture.

## C  Impact of the Multi-Head Self-Attention

We present in table 1 the impact of the number of heads in the self-attention mechanism on performance. Using multiple heads is clearly beneficial for the model's performance. This is consistent with the notion that multi-heads enable richer interpretations of the sequence by separating sections of the embedding and learning different aspects of the interplay between related tokens [9].

Table 1: A comparison of the negative natural logarithm of Bit Error Rate (BER) for three normalized SNR values (4,5,6) of our method for different number of self-attention heads $h$. Higher is better. All the models are $N = 6, d = 32$ ECCT.

| Method | $h = 1$ | | | $h = 4$ | | | $h = 8$ | | |
|---|---|---|---|---|---|---|---|---|---|
| | 4 | 5 | 6 | 4 | 5 | 6 | 4 | 5 | 6 |
| Polar(64,32) | 5.33 | 7.01 | 9.06 | 5.65 | 7.49 | 9.89 | 5.71 | 7.63 | 9.94 |
| LDPC(49,24) | 5.44 | 7.62 | 10.70 | 5.69 | 7.96 | 11.21 | 5.74 | 8.13 | 11.30 |
| BCH(63,36) | 4.33 | 5.76 | 7.79 | 4.44 | 5.94 | 8.11 | 4.42 | 5.91 | 8.01 |

## D  Comparison with Successive Cancellation List (SCL) Polar Decoder

Table 2 compares the performance of our model to the SOTA SCL Polar decoder [8] for several Polar Codes. The SCL decoder has a time and space complexity of $\mathcal{O}(LN \log N)$ and $\mathcal{O}(LN)$, respectively. We tested the SCL algorithm for $L = \{1, 4\}$ and we sampled *only* $10^5$ noisy codewords because of the high complexity and non-parallel application of the SCL algorithm. The proposed *shallow* ECCTs are able to compete and even surpass the SCL for some of the codes and SNRs. Increasing the capacity of the network, especially with more layers, is expected to lead to better results as demonstrated for LDPC codes. Similarly, SCL with bigger lists would obtain improved accuracy.

## E  Non Gaussian Channel

In this Section, we test our framework on a non-gaussian Rayleigh fading channels, whcih are often used for simulating the propagation environment of a signal, e.g., for wireless devices.

In this fading model, the transmission of the codeword $x \in \{0, 1\}^n$ is defined as $y = hx_s + z$, where $h$ is an $n$-dimensional i.i.d. Rayleigh distributed vector with a scale parameter $\alpha$, and $z \sim \mathcal{N}(0, \sigma^2 I_n)$.

In our simulations, we assume a *high* scale $\alpha = 1$ in order to easily compare and reproduce the results, while the level of the Gaussian noise and of the testing procedure remains exactly the same as described in the paper. The overall variance of the transmitted codeword $y$ in the Rayleigh channel is roughly *twice* the AWGN's on the tested SNR range.

Table 2: A comparison of the negative natural logarithm of Bit Error Rate (BER) for three normalized SNR values (4,5,6) between the proposed method with $N = 6, d = 128$ and the SOTA SC-L algorithm. Higher is better.

| Method | SC-$L=1$ | | | SC-$L=4$ | | | ECCT | | |
|---|---|---|---|---|---|---|---|---|---|
| | 4 | 5 | 6 | 4 | 5 | 6 | 4 | 5 | 6 |
| Polar(64,32) | 7.28 | 9.84 | 12.50 | 7.55 | 10.21 | 13.60 | 6.99 | 9.44 | 12.32 |
| Polar(64,48) | 6.16 | 8.22 | 11.30 | 6.17 | 8.22 | 11.30 | 6.36 | 8.46 | 11.09 |
| Polar(128,64) | 8.42 | 11.10 | - | 8.46 | 11.15 | - | 5.92 | 8.64 | 12.18 |
| Polar(128,86) | 7.36 | 9.76 | 12.40 | 7.37 | 9.76 | 12.45 | 6.31 | 9.01 | 12.45 |
| Polar(128,96) | 6.78 | 9.13 | 11.92 | 6.80 | 9.14 | 12.01 | 6.31 | 9.12 | 12.47 |

The results are presented in Figure 4. As can be observed, our method is still able to learn to decode even under these very noisy fading channels.

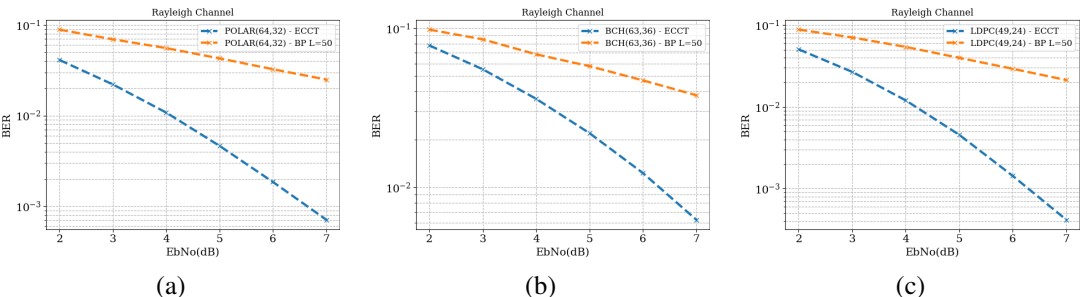

(a)          (b)          (c)

Figure 4: BER comparison between BP and the proposed ECCT $N = 6, d = 32$ for the Rayleigh fading channel for various values of SNR for (a) Polar(64,32), (b) BCH(63,36), and (c) LDPC(49,24) codes.

## F  Robustness to the Modulation

In this Section, we test our framework on a modulation other than the Binary Phase-shift keying. The 16 Quadrature amplitude modulation (16QAM) is common for wireless fading. We simulate the Bit-Interleaved Coded Modulation (BICM) demapping [11, 2] as follows

$$y = Demap\big(Map(x) + z\big), \tag{1}$$

where $Map$ and $Demap$ represent the 4-bits per symbol constellation (16QAM) mapping and demapping, respectively, and $z$ denotes the $n/4$-dimensional *complex* AWGN noise, with noise levels that are defined using the same protocol as in our main experiments. This protocol has been implemented using NVIDIA's Sionna library [3, 1].

The results are presented in Figure 5. Evidently, our model is able to learn accurate decoding under different modulation even on low SNRs.

## G  Comparison with Augmented Neural BP

In this Section, we compare the complexity and the performance of ECCT with the complexity of the Neural BP (NBP) [4], which does not suffer from the high computational cost of the later hyper-network based models [5].

We recall that the time complexity of the (neural) BP is defined as $\mathcal{O}(L\big(\sum_i^c d_i^2 + \sum_i^v \tilde{d}_i\big))$, where $d_i$ denotes the degree of the $i$-th parity check node, $\tilde{d}_i$ denotes the degree of the $i$-th variables nodes, and $L$ is the number of iterations.

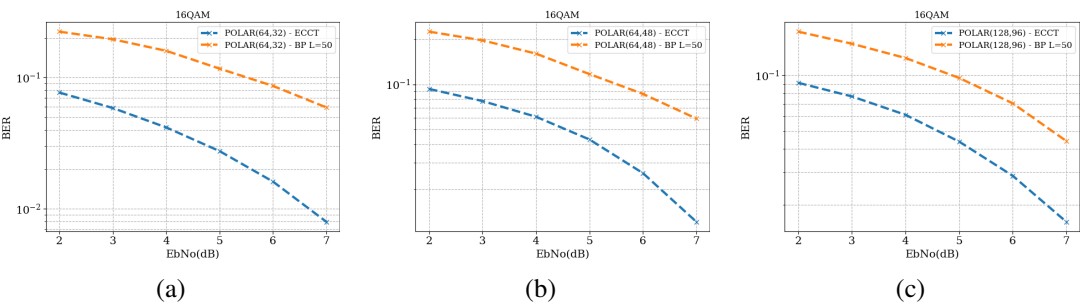

(a)            (b)            (c)

Figure 5: BER comparison between BP and the proposed ECCT $N = 6, d = 32$ under 16QAM modulation for various values of SNR for (a) Polar(64,32), (b) Polar(64,48), and (c) Polar(128,96) codes.

We present in Table 3 the numerical time complexity values of different ECCTs and the neural BP decoder. The (neural) BP is obtained at convergence with $L = 50$ as reported in [5]. These results are the best results that the neural BP architecture can reach. The inability of the (neural) BP to further improve, was the motivation for the development of more computationally expensive augmented BP methods [5, 6]

The shallower ECCTs (i.e. $N = 2, d = 32$) are comparable to the complexity of BP for most of the codes and is even much more efficient for several codes such as BCH codes (the level of sparsity of each code is the major cause of the difference in complexity). For LDPC codes, for which BP is specially fitted and reliable [7], BP's complexity is much lower than ours. However, as described in Section 6.3, many acceleration methods can greatly improve the complexity. For example, low-rank approximations [10] can reduce the quadratic complexity in $d$ to *linear*.

In the current setting, one can observe that, at similar complexity, $N = 2$ ECCTs are too shallow and thus cannot always reach NBP's accuracies. The most impactful architectural parameter on the reliability of the ECCT is the number of layers $N$, while the embedding size $d$ has the most influence on the complexity (quadratic). Thus, one can compare, for example, the results for the BCH(63,45) code, for which BP has a similar complexity to the $N = 6, \boldsymbol{d = 16}$ ECCT. The negative logarithm of the BER for $E_b/N_0 \in \{4, 5, 6\}$ is $4.98, 6.77, 9.26$, respectively, for ECCT, while the neural-BP of [4] reaches at convergence $4.49, 6.01, 8.20$ [5], a 12% improvement on average. Similarly, on the Polar(128,96) code, the accuracy of the $N = 6, \boldsymbol{d = 12}, \boldsymbol{h = 4}$ ECCT is $4.98, 6.85, 9.25$ while NBP reaches $4.63, 6.31, 8.54$, an 8% improvement on average.

Table 3: Numerical comparison of the complexity of the (Neural) BP of [4] and of the proposed ECCT for different $N$ and $d = \{32, 64, 128\}$.

| Method | NBP[4] | ECCT N=2 | | | ECCT N=6 | | |
|---|---|---|---|---|---|---|---|
| | 50 | 32 | 64 | 128 | 32 | 64 | 128 |
| POLAR(64,32) | 231600 | 638336 | 1669888 | 4912640 | 1915008 | 5009664 | 14737920 |
| POLAR(64,48) | 244600 | 388864 | 1105408 | 3521536 | 1166592 | 3316224 | 10564608 |
| POLAR(128,64) | 1074400 | 2236672 | 5259776 | 13665280 | 6710016 | 15779328 | 40995840 |
| POLAR(128,86) | 1305200 | 1503104 | 3702528 | 10190336 | 4509312 | 11107584 | 30571008 |
| POLAR(128,96) | 1244800 | 1245184 | 3145728 | 8912896 | 3735552 | 9437184 | 26738688 |
| LDPC(49,24) | 78400 | 407680 | 1118464 | 3449344 | 1223040 | 3355392 | 10348032 |
| LDPC(121,60) | 435600 | 1965696 | 4676864 | 12335616 | 5897088 | 14030592 | 37006848 |
| LDPC(121,70) | 363000 | 1803776 | 4312064 | 11442176 | 5411328 | 12936192 | 34326528 |
| LDPC(121,80) | 290400 | 1654656 | 3972864 | 10599936 | 4963968 | 11918592 | 31799808 |
| MACKAY(96,48) | 100800 | 1483776 | 3557376 | 9474048 | 4451328 | 10672128 | 28422144 |
| CCSDS(128,64) | 230400 | 2445312 | 5677056 | 14499840 | 7335936 | 17031168 | 43499520 |
| BCH(63,36) | 533900 | 486144 | 1340928 | 4156416 | 1458432 | 4022784 | 12469248 |
| BCH(63,45) | 413600 | 362880 | 1057536 | 3442176 | 1088640 | 3172608 | 10326528 |
| BCH(63,51) | 501000 | 262144 | 831488 | 2891776 | 786432 | 2494464 | 8675328 |

## H  Performance on Larger Codes

We present in Figure6 the performance of the proposed ECCT on larger codes.

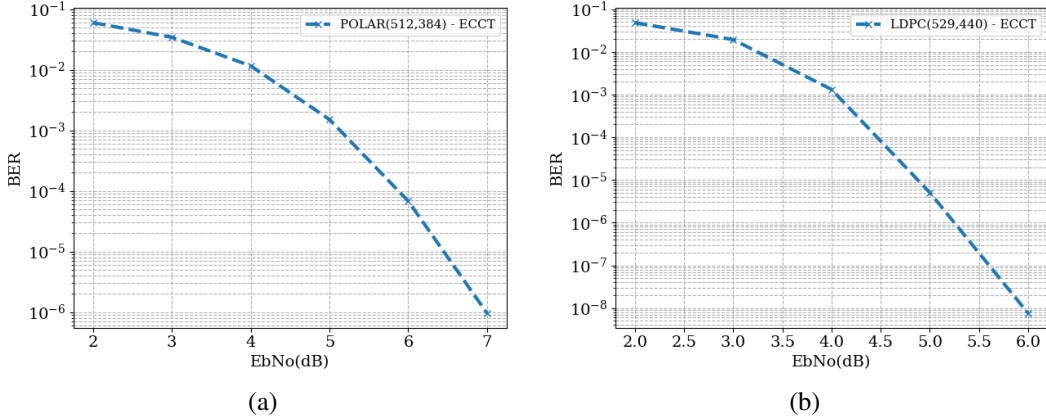

(a)

(b)

Figure 6: ECCT $N = 6, d = 32$ performance for various values of normalized SNR for (a) Polar(512,384), (b) LDPC (529,440) codes.