# OpenReview forum: "Error Correction Code Transformer"
_NeurIPS.cc/2022/Conference — NeurIPS 2022 Accept_

### Official Review · Reviewer_MQDw · 2022-07-10

**Rating:** 7
**Confidence:** 3
**Soundness:** 3 good
**Presentation:** 4 excellent
**Contribution:** 3 good

**Summary:**

This paper studies neural decoding of (linearly coded) transmissions in the physical communication layer. It proposes a novel architecture that has three main features: it is based on transformers, contains a scaled element-wise embedding of the input, and has an adapted-mask self-attention mechanism. The proposed architecture is compared empirically to existing neural decoders and is shown to outperform them with higher decoding success and less computational complexity.

**Questions:**

Question 1: in related work, a dichotomy between model-based on model-free was presented and the authors of the present work place theirs’ in the model-free category. Since the present work uses expert knowledge in how the adapted-mask is generated, one could argue that it belongs somewhere in the middle of these categories. Could the authors elaborate on this?

Question 2: Would there be a way to compare neural decoders to other non-learned SOTA decoders? Would it be possible to do more quantitive comparison as well?

**Limitations:**

See above.

**Strengths And Weaknesses:**

Strengths:
----------------

Strength 1: The presented method is the first transformer-based decoder and it also gives an efficient architecture.

Strength 2: The proposed method has better decoding performance than other neural network based approaches.

Strength 3: The proposed method allows to input expert knowledge to the system and has lower complexity than other neural network based methods.


Weaknesses:
---------------------

Minor Weakness 1: Only AWGN channel is considered in this work. This may be in line with other neural decoder works, but it would be important to consider also wireless channels with fast fading and e.g., with interference, which change the noise statistics. There is a possibility that the learned methods might provide even further gains in these scenarios by learning the noise correlation statistics.

Minor Weakness 2: Especially with wireless fast fading channels, other modulations than BPSK should be considered, at least 16QAM.

After the author feedback (rebuttal):
------------------------------------------
I have reviewed the author feedback and they have done good job in clarifying most of the questions the reviewers have posed. I have changed the score accordingly.

---

> ### Author Response · Authors · 2022-08-01
> **Authors response**
>
> Thank you for the supportive and comprehensive review.
>
> ## Model-based vs. model-free
> We agree the terminology can be misleading since our Transformer network uses a code-based mask (we also provide the unmasked Transformer as an ablation study).
>
> The terminology has been borrowed from [26] and refers to decoders that implement any variants of generic types of neural networks. This contrasts with the model-based methods, in which an existing non-neural decoder (e.g. BP) is augmented with learnable parameters. We clarified this in the revised version.
>
> ## Comparison to SOTA non-neural decoders
> In Appendix D we add, besides BP, the performance of the SC-L for polar codes. As can be seen,  relatively shallow ECCTs can compete and sometimes even surpass the SCL for some of the codes and SNRs. Increasing the capacity of the network, which currently has only six layers, may further improve the performance to set new SOTA results.
>
> A rigorous comparison should take into account exact complexity analysis as well as the potential acceleration of the ECCT as suggested in Section 6.3. For example, a low-rank approximation, e.g. Linformer [32], would transform the quadratic complexity in $d$ to linear, which could make ECCT extremely competitive on the algorithmic complexity level as well.
>
> ## Non-Gaussian channel
> In Appendix E, we present the performance of our model for a Rayleigh channel, where we can observe the ECCT remains effective even for such channels.
>
> ## 16QAModulation
> In Appendix F we add the performance under 16QAModulation. ECCT’s advantage is maintained under different modulations.

---

> > ### Comment · Reviewer_MQDw · 2022-08-08
> > **Thank you**
> >
> > Thank you for the reply and the revised manuscript. I have read them and adjusted the score accordingly.

---

### Official Review · Reviewer_Nt2o · 2022-07-11

**Rating:** 7
**Confidence:** 4
**Soundness:** 3 good
**Presentation:** 3 good
**Contribution:** 3 good

**Summary:**

This paper considers a very important and interesting topic: the decoding of error-correcting codes.
Over the past few years, there has been a lot of work on applying learning to the decoding problem. Various neural architectures have been considered as well, starting from feedforward networks to recurrent and convolutional neural networks. There also have been a long line of work which introduced learnable parameters to the existing decoder architecture, some of which are considered as a baseline in this paper.

There has not been a transformer-based channel decoder yet; training a transformer that can successfully decode error-correcting codes is empirically quite challenging. This paper addresses the challenge by utilizing the knowledge of codes in a way; using the appropriate masking and modulating the embedding by syndrome values. This approach is novel and interesting - so I would like to advocate the acceptance of this paper in that regard.


**Questions:**

I'd be curious to see the comparison between the reliability of neural-augmented BP and the transformer  (vanilla, no weight adaptation network) -based decoder when their complexities are similar.

I'd be also curious to see the performance of these decoders for non-Gaussian channels. Is there a difference in terms of robustness?

**Limitations:**

The authors mention that the complexity could be potentially improved by several techniques.

**Strengths And Weaknesses:**


(Strength)
As mentioned in detail above, the paper successfully demonstrates a transformer-based channel decoder for the first time.

(Weakness)
Despite the idea being novel and the results being promising, I have two major concerns.

The first is whether the comparison between the BP families and the transformer families is fair. Given that the transformer-based decoder is not a purely data-driven decoder, we would expect to see gains compared to the existing decoder. For example, L-layered augmented BP decoders typically have complexity similar to the L-layered BP decoders. Hence, the complexity of the augmented BP decoder is similar to the complexity of the traditional BP decoder but is superior in terms of reliability. [Hyper-BP might be computationally expensive due to the 'hyper network' which selects the weights. However, there are other neural augmented BP algorithms that do not include the weight selection module but are pretty good.]

Second, the presentation and ablation studies can be further improved. For example, section 6.1 and the description of Figure 5 lack details. Also, I wonder if any similar ablation study can be done for the polar or LDPC or BCH codes. Section 6.2 is interesting and insightful. In Section 3, providing the number of operations would be more informative (and potentially mention vanilla augmented-BP algorithms as well).

For these reasons, my current score is 4, but I am really at the exact borderline. I would like to listen to the authors; I will go through the rebuttal and fix my score afterwards.

---
After the rebuttal: The new experiment results (e.g., Appendix G) are exactly are very informative and satisfactory. The reliability gap between the transformer-based decoders and neural BP decoders does not seem huge. (It is a bit hard to interpret the gain from the negative logarithm of the BER, and the SNR gain might be more indicative.) Nevertheless, I still really like the idea of utilizing the transformer architecture for decoding, and the updated manuscript is comprehensive. I'd updated my score accordingly.

---

> ### Author Response · Authors · 2022-08-01
> **Authors response**
>
> We thank the reviewer for the mostly supportive review and detailed feedback.
>
> We note that despite writing “*I would like to advocate the acceptance of this paper in that regard*”, the overall grade was slightly lower than the acceptance threshold. We kindly ask to know if our answers satisfy all of the concerns raised, and if not, we would be happy to future comply.
>
> ## Section 6.1. experiments
> We added clarifications in this section, and, following the review, the revised manuscript provides in Appendix B another illustration for a larger BCH code.
>
> ## Comparison with vanilla augmented BP
> We now also provide in Appendix G the complexity of vanilla augmented-BP algorithms as well as numerical simulations of the complexity for different codes. We analyze and compare the complexities and performance, and also provide the performance of two models applied on two different codes, that have similar complexity to the ‘at convergence’ neural BP decoders. These models improve upon Neural BP by 12\% and 8\% on average over the normalized SNR range.
>
> We note that the vanilla augmented-BP has reached its full capacity and is not able to further improve its performance, contrary to the proposed ECCT.
> Also, computational complexity may not be the only appropriate metric for efficiency. Our current implementation (even for 6 layers) is much faster than a 100 layers/50-iterations (neural or not) BP even on a general purpose GPU, and can provide much shorter latencies and higher throughput. Furthermore, the complexity of Transformers can be greatly reduced via the vast amount of recently developed methods as mentioned in Section 6.3. For example, a low-rank approximation, e.g. Linformer [32], would transform the quadratic complexity in $d$ to linear, which could make ECCT extremely competitive on the algorithmic complexity level as well.
>
> ## Performance on a non-Gaussian channel.
> Following the review, we provide in Appendix E a comparison between BP and our method for a Rayleigh channel. ECCT’s advantage is maintained in such channels.

---

### Official Review · Reviewer_AayN · 2022-07-16

**Rating:** 7
**Confidence:** 4
**Soundness:** 4 excellent
**Presentation:** 3 good
**Contribution:** 3 good

**Summary:**

- The paper presents a novel Transformer based generic decoding procedure for typical linear error correction code families (LDPC, Polar etc..).
- The method does achieve SOTA performance on mid block length codes (~100-200)
- The key novel contribution of the work was the introduction of positional reliability embeddings and the attention mask, both of which (particularly the mask) uses some clever information theoretic domain understanding. These ideas improve the convergence and hence the performance of the model
- One key selling point of the paper is that the method is general enough to be applied to any linear code without any modification.

**Questions:**

- The use of mask is quite nice, and the authors claim that it reduces the complexity by 84%.. does this reduction in complexity lead to reduction in FLOPS/run-time.. or just a symbolic reduction which leads to lower complexity and better convergence? Please clarify

- The intuition behind positional reliability encoding is intuitively clear to me, at least for the elements corresponding to |y_i|.. can the authos explain the intuition behind the encoding for the syndrome bits?

- Comparison with List-SC decoding for polar codes might be interesting.

**Limitations:**

No societal limitations

**Strengths And Weaknesses:**

Strengths:
- The paper presents a novel way of using transformers and self attention to achieve strong results in terms of the BER
- The architecture isn't just using a ML-model and blindly using it for the decoding problem. The training uses some clever ideas from the code construction to improve the convergence.

Weaknesses:
- The explanation was overall good, but in some places a bit lacking. More detailed comments at the end
- The authors briefly mention that the runtime/power numbers might be sub-par as compared with non-ML methods.. still it might be great to see the comparison to know where the field is at.

Specific suggestions:
- A couple of typos.. line 117 propriety -> property
- $y_b$ -> although clear, it is not defined in the manuscript
- The postprocessing an pre-processing is not clear.. please describe more clearly. For example: it is not clear to me (without reading the cited reference, what you mean by "The post-processing step plugs back the vector elements of y...."
- The use of mask is quite nice, and the authors claim that it reduces the complexity by 84%.. does this reduction in complexity lead to reduction in FLOPS/run-time.. or just a symbolic reduction which leads to lower complexity and better convergence? Please clarify

---

> ### Author Response · Authors · 2022-08-01
> **Authors response**
>
> Thank you for the very supportive and comprehensive review.
> We thank the reviewer for pointing us to important corrections and typos. All have been addressed in the revisited manuscript.
>
> ## The effect of masking on complexity
> The proposed masking approach can lead to a great reduction of the run-time and power on dedicated devices via an adapted memory fetching and by processing paired elements only in the tensor/matrix multiplication unit (e.g. https://arxiv.org/ftp/arxiv/papers/1704/1704.04760.pdf).
> Besides the fact the mask is very sparse, since the mask is symmetric and the dot-product is a bilinear symmetric operation (over real numbers), only half of the computations are required.
> The current implementation employs a general-purpose GPU, simulating the masking effect. The main goal of the experiments is to demonstrate the effect of masking on accuracy.
>
> ## An intuition for encoding the syndrome bits
>  A non-zero syndrome means that at least one particular parity-check bit would give a negative value via the binary to sign mapping $f(s_{i})=1-2s_{i}$. This sign is modulated by its magnitude $|y_i|f(s_{i})$ ensuring the reliability of this same parity check bit. Thus, a non-zero parity-check bit is easily detectable, and its contribution is diminished via the softmax self-attention mechanism (exponent of a negative number).
>
> ## Comparison with List-SC decoders for Polar codes
> Appendix D of the revised manuscript contains comparisons with the linearithmic SOTA SC-L decoder for all the Polar codes used in our experiments. As can be observed, the proposed *shallow* ECCTs can compete and sometimes even surpass the SCL for some of the codes and SNRs. Increasing the capacity of the network, which currently has only six layers, should further improve the performance as with LDPC codes.

---

### Official Review · Reviewer_FPkL · 2022-07-18

**Rating:** 6
**Confidence:** 4
**Soundness:** 2 fair
**Presentation:** 2 fair
**Contribution:** 2 fair

**Summary:**

This paper proposes a Transformer based that employs relaxed inductive bias compared tgo Tanner graph-based methods while utilizing domain knowledge compared to standard model-free decoders based on fully-connected graph.


**Questions:**

- Figure 3: Is the multi-head self-attention block still necessary? Given that we’re looking for the localized bits that meet each row of the parity check (PC) matrix, will single-head self-attention suffice? In other words, if you permute the rows of the PC matrix, you will get the same result, indicating that the problem has some sort of equivariance property that the authors may utilize in the Transformer design

- Figure 2: This is a good illustration; it would be fascinating to see how the attention maps visualized after the training and how they compare to the restrictive Tanner graph; it would also be interesting to see if the mask-based decoders genuinely help attention to more targeted interactions or not

- Ln 147: Meant Algorithm 1?


**Limitations:**

- Ln 154-157: Is the complexity reduction in 4.2 from code-aware self-attention adequate enough to apply the method to longer LDPC codes, such as greater than 512, say 1K or 4K lengths? If it’s not, the authors should have stated that clearly as a limitation or future work of the study

**Strengths And Weaknesses:**

This paper's main strength is the well-defined scope of the problem.

The paper's main weakness is that the proposal’s benefits were not studied thoroughly. For examples, I have a few questions below

---

> ### Author Response · Authors · 2022-08-01
> **Authors response**
>
> Thank you for the supportive and comprehensive review.
>
> ## The need for multi-head attention
> The Transformers are indeed permutation equivariant models. However, the multi-head attention aims at enriching the analysis of the embedding and not of the elements. We now provide in Appendix C experimental results regarding the impact of the number of heads. As can be observed, using more than one head is beneficial for performance.
>
> ## Additional illustrations of self-attention maps
> We now provide in Appendix A the illustration of several self-attention maps for different codes with their corresponding inputs. Interestingly, we can observe the ECCT seems to focus its processing on the syndrome in the early stage.
>
> ## Typo in line 147
> We thank the reviewer for the correction. This typo was due to the wrong placement of the \label.
>
> ## Larger LDPC codes
> The method can be applied at arbitrary code length under potentially high memory and computational training constraints. For example, running our code on a Polar(512,384) code requires 7x more time per epoch, which is computationally intensive but still feasible. In other domains, Transformers are often run on 512-1024 tokens, which supports the viability of our method for larger codes.
> Also, we established in our experiments that our framework is much more scalable than the classical networks used by [2], which struggle with learning larger codes such as $n=127$ (Figure 4.c). Following the review, we provide in Appendix H the performance of the ECCT on two larger codes. We can observe that ECCT can learn to efficiently decode larger codes as well.

---

> ### Author Response · Authors · 2022-08-09
> **We would be happy to discuss any remaining concerns**
>
> Thank you again for the valuable ideas, which have no doubt helped improve our manuscript.
> We would be happy to know if you are satisfied with our answers, or if there is anything else we can address.

---

### Author Response · Authors · 2022-08-01
**Summary of changes**

We have uploaded a revised version of our manuscript, which contains the recommended clarifications and additional results specifically requested by the reviewers.

Following a request by FPkL, we provide in appendix A illustrations of self-attention maps for several codes.

Following a request by FPkL, we have added in appendix C experiments assessing the impact on the accuracy of the number of heads in the self-attention layers.

Following a request by FPkL, we have added in appendix H the performance of our model for an LDPC code with $n=529$ and a Polar code with $n=512$.

Following a request by AayN and MQDw, we have added in appendix D the performance of the SCL decoder for Polar codes.

Following a request by Nt2o, we have added in appendix B an ablation validating the impact of the reliability embedding. It employs a larger BCH code.

Following a request by Nt2o, we provide in Appendix G the complexity of BP, numerical simulations of the complexity, and accuracy comparisons between our shallower model and a neural BP model that has a similar complexity.

Following a request by Nt2o and MQDw, we provide in Appendix E experiments of our method with a non-Gaussian (Rayleigh) channel.

Following a request by MQDw, we have added in Appendix F experiments of our method with 16QAModulation.

---

### Author Response · Authors · 2022-08-08
**We would be happy to address any follow-up questions**

We appreciate the reviewers' detailed comments and valuable suggestions. We have made an effort to factually address the stated issues, as indicated in the summary of changes that has been posted.

If the response to each reviewer has not already addressed all concerns, we would appreciate the opportunity to further discuss our work.

---

### Meta-Review · Area_Chair_a8kw · 2022-08-20

**Recommendation:** Accept
**Confidence:** Certain

**Metareview:**

This paper is part of a popular line of research aiming to apply neural network concepts to the decoding of error-correcting codes. The main novelty consists in the introduction of an architecture based on transformers. The authors provide convincing and thorough numerical results comparing the BER and the complexity of the proposed approach with various baselines. Such results apply to codes in the short to medium block-length range (from 32 to 128 bits).

The reviewers have expressed a number of concerns in their initial reports. After the rebuttal stage, most of these concerns have been resolved. The reviewers Nt2o and MQDw have particularly appreciated the additional numerical results provided by the authors (BP baselines, non-Gaussian channels, other modulations and SCL decoder for polar codes). This is also explicitly pointed out in the updated reviews.

In summary, there is clear consensus towards accepting the paper.  After my own reading of the manuscript, I agree with this assessment and I am happy to recommend acceptance. As a final note, I would like to encourage the authors to include in the camera ready the additional experiments and discussions mentioned in the rebuttal.


**Award:**

No

---

### Decision · Program_Chairs · 2022-09-14

Accept